# Green synthesized *Polyscias fulva* silver nanoparticles ameliorate uterine fibroids in female Wistar Albino rats

Kenedy Kiyimba[1,2,3], Ayaz Ahmed[4*], Muhammad Iqbal Choudhary[5], Khadija Rehman[5], Syed Mehmood Hasan[6], Abdul Jabbar[5], Samuel Baker Obakiro[2,3], Muhammad Raza Shah[5], Were Lincoln Munyendo[7], Eric M. Guantai[1], Yahaya Gavamukulya[8]

**1** Department of Pharmacology and Pharmacognosy, School of Pharmacy, University of Nairobi, Nairobi, Kenya, **2** Natural Products Research and Innovation Centre, Busitema University, Mbale, Uganda, **3** Department of Pharmacology and Therapeutics, Faculty of Health Sciences, Busitema University, Mbale, Uganda, **4** Dr. Panjwani Center for Molecular Medicine and Drug Research (PCMD), International Center for Chemical and Biological Sciences, University of Karachi, Karachi, Pakistan, **5** H.E.J Research Institute of Chemistry, International Center for Chemical and Biological Sciences, University of Karachi, Karachi, Pakistan, **6** Department of Pathology, Jinnah Sindh Medical University, Karachi, Pakistan, **7** School of Pharmacy & Health Sciences, United States of America International University-Africa, Nairobi, Kenya, **8** Department of Biochemistry and Molecular Biology, Faculty of Health Sciences, Busitema University, Mbale, Uganda

* ayaz.ahmed@iccs.edu

## Abstract

Uterine fibroids affect a substantial proportion of women in their reproductive age. Despite their effectiveness, surgical options such as hysterectomy are invasive, costly, and associated with recurrences. Pharmacological treatments are non-curative, only alleviate symptoms, and associated with adverse effects. *Polyscias fulva (Araliaceae)* is traditionally used to manage uterine fibroids in East Africa. In this study we synthesized *Polyscias fulva* silver nanoparticles (PFAgNPs), evaluated their toxicity and activity against monosodium glutamate (MSG)-induced uterine fibroids in Wistar albino rats. The UV-visible spectroscopy showed maximal absorbance at 425 nm with adequate stability at varying temperatures, pH and storage conditions. Dynamic light scattering (DLS) analysis revealed an average hydrodynamic size of 107.4 d.nm, polydispersity index of 0.264, and zeta potential of -18.3 mV. X-ray diffraction (XRD) confirmed the crystalline nature of PFAgNPs with an average size of 25 nm while scanning electron microscopy (SEM) showed a spherical shape with an average size of 35 nm. The PFAgNPs caused lethargy, hyperventilation, and hyperactivity at a dose of 300 mg/kg BW, whereas 2000 mg/kg caused severe toxicity, resulting in death in acute toxicity testing. The no observed adverse effect level was 50 mg/kgBW, the lowest observed adverse effect level was 100 mg/kgBW, and median lethal dose ($LD_{50}$) was 1000 mg/kg. The PFAgNPs significantly decreased ($P < 0.05$) serum proteins, cholesterol, estrogen and progesterone alongside preservation of the

**Data availability statement:** All data generated or analyzed during this study is available on this link: https://doi.org/10.6019/S-BSST1912.

**Funding:** Funding This study was funded by COMSTECH-NAPRECCA Consortium Through a PhD Fellowship (award number: 3(250)/23-COMSTECH) awarded to KK at the International center for Chemical and Biological Sciences (ICCBS), University of Karachi, Pakistan. The funders had no role in study design, data collection and analysis, decision to publish, or preparation of the manuscript.

**Competing interests:** The authors have declared that no competing interests exist.

histoarchitecture of the uterus. Further research is needed to investigate the clinical safety of PFAgNPs in managing uterine fibroids.

## Introduction

Approximately 70% of women in their reproductive age are affected by uterine leiomyomas before menopause [1]. In over 25% of the cases, the clinical symptoms are severe enough to require treatment [2]. The neoplasms are associated with a number of gynecological complications, such as menorrhagia, intermenstrual bleeding, and increased risks during pregnancy, including miscarriage and premature birth [3]. While surgical interventions like hysterectomy and myomectomy are the most definitive treatments, they are invasive, costly, and associated with a high recurrence rate [4]. Moreover, the pharmacological agents such as the centrally acting gonadotrophin releasing analogs (cetrorelix and leuprolide), the peripherally acting drugs (aromatase inhibitors, antiprogestins), and the selective progesterone receptor modulators [5] have limited efficacy and are often associated with several adverse drug reactions [6,7]. This calls for the development of newer, safe yet effective therapeutic approaches for management of uterine fibroids.

*Polyscias fulva (*Araliaceae) is a deciduous tree used in East and Central Africa to manage conditions such as cancer [8], benign tumors [9], venereal, bacterial and dermatophyte infections [10]. The plant has demonstrated several pharmacological properties including cytotoxic, antibacterial and anti-dermatophytic effects [8,10,11]. These effects have been attributed to some of its phytochemical compounds which include; phenolics, steroids, triterpenes, terpenoids, and saponins [12].

The pharmaceutical industry has recently gained interest in nanoparticle formulation for therapeutic modalities and drug delivery [13]. Their nano size(1–100nm), good solubility, surface functionalization, and multi-functionality, present them as ideal therapeutic options in oncology. Their enhanced permeability and high retention effects enable precise tumor targeting hence offering prolonged effects within the tumor microenvironment [14]. Particularly, the silver nanoparticles (AgNPs) as compared to the other metallic nanoparticles, have gained considerable attention for their safety profile and potential pharmacological effects against pathogenic bacteria, tumor cells, inflammation [15–17].

While the conventional methods for synthesizing metallic nanoparticles are associated with high production costs and pose environmental and biological toxicity risks [18,19], The green synthesis approach is not only an eco-friendly but also affordable and safer alternative, utilizing plant phytocompounds as reducing and stabilizing agents. The bioactive compounds within plants serve as reducing agents to produce metallic nanoparticles with good biological activities, while minimizing the production of toxic byproducts [20]. Due to its rich phytochemical composition, *Polyscias fulva* is a suitable plant for the synthesis of AgNPs, as these could facilitate the formation of the nanoparticles while enhancing their bioactivity and potentially reducing their toxicity.

The nanoparticle formulation of this plant has never been studied to cure uterine fibroid models. So, Green synthesized *Polyscias fulva* silver nanoparticles

(PFAgNPs) could potentially offer a novel alternative treatment modality. The combination of the plant's inherent pharmacological properties with the enhanced AgNPs efficacy could pave the way for several therapeutic applications in drug delivery systems, tissue regeneration, and potential cancer therapies, further broadening the scope of these AgNPs in modern medicine.

This current study presents the green synthesis approach for PFAgNPs using *Polyscias fulva* stem bark aqueous extract, their Physical chemical characteristics and acute toxicity effects in Wistar albino rats. Furthermore, their ameliorative potential against an experimental animal uterine fibroid model in female Wistar albino rats was also evaluated. This study seeks to establish the PFAgNPs as a novel, non-invasive treatment alternative for uterine fibroids, with the potential for broader applications in biomedical research.

## Methods

### Plant sample collection

The *Polyscias fulva* stem barks were harvested from Mabira Forest, Buikwe district Central Uganda (0°24'11.6"N 33°00'22.6"E) in June 2023 with the guidance of a taxonomist. Permission to access the field sites was granted by the local area administration, as the plant specimens were collected from individual home gardens rather than from a protected forest area. The plant samples were transported in an airtight bag to the pharmacology laboratory at the Faculty of Health Sciences, Natural Products Research and Innovation Centre, Busitema University for further processing. The stem barks were sliced into small pieces and kept for 25 days under a shade to air dry. The dried samples were pulverized using a Flashpoint BU01 machine, and the powder was kept at room temperature in airtight envelopes. A voucher specimen (KK_/001_/2023) was deposited in the herbarium at the Department of Plant Sciences, Microbiology & Biotechnology, Makerere University.

### Plant sample preparation

A total of 100g of the dry stem-bark powder of *Polyscias fulva* was macerated in 500mL of distilled water and the preparation placed in a shaking water bath and heated to 60°C for 30 minutes. The preparation was then allowed to cool, filtered with a filter paper (Whatman No. 1), placed into 50mL Erlenmeyer flasks, labeled as *Polyscias fulva* crude aqueous extract (PFE) and then stored in a refrigerator(4–8°C) for further use [21].

### Green synthesis of PFAgNPs using *Polyscias fulva* Extract (PFE)

Five(5mL) of PFE was mixed with about 50ml of 1mM AgNO3 solution in a 100 ml conical flask and mixed thoroughly, forming a uniform mixture. After about 2 hours, an observable color change of the reaction mixture was recorded. After 24hrs of incubation at 24 h at room temperature, the color of the reaction mixture had completely changed from dark brown to yellowish brown indicating the successful synthesis of PFAgNPs[22]. All solutions were freshly prepared. The resultant solution was centrifuged at 6000rpm for 10 minutes using an Ortoalresa, Biocen 22 R centrifuge and dried in air. The air-dried pellets were used for further characterization by UV–Vis, DLS, AFM, SEM, EDX, FTIR and XRD analysis.

### Physicochemical characterization of the PFAgNPs

The UV/Visible Scanning Spectrophotometer(Shimadzu UV-1800) was used to record the spectra of the synthesized PFAgNPs[23]. The synthesized PFAgNPs were exposed to varying temperatures (25, 35, 45, 55, 65, 75, and 85°C), whereas for pH stability, the nanoparticles were exposed to varying pH (2, 4, 7, 9, and 11) conditions and their absorbance spectra recorded in a scan range of 200–800 nm [24,25]. Their polydispersity index, zeta potential and hydrodynamic size was determined by the Zetasizer instrument (ZS-90 Malvern Instruments UK). The surface topography of the synthesized PFAgNPs was visualized by the Atomic force microscopy (Agilent 5500); appropriate dilutions of the PFAgNPs preparation

were made, and then one drop of the suspension was put on the mica slide and air dried and then analyzed by AFM [22]. To investigate the shape, size and morphology of the PFAgNPs, the scanning electron microscope/Energy dispersive X-ray Spectrometer system (FEI XL30 Sirion FEG, Oxford Instruments Plc, Abingdon, UK) was used. The SEM system was operated at 6 kV accelerating voltage, whereas the EDX system had an EDAX lithium-doped silicon semiconductor detector for elemental analysis. The functional groups in the PFAgNPs and PFE were determined by Functional group analysis using the Fourier Transform Infrared Spectroscopy (Thermo scientific Nicolet iS50) following the potassium bromide (KBr) disk method. The size of the PFAgNPs were determined by the X-ray diffraction (XRD) machine (D8 Advance; Bruker Optik, Ettlingen Germany). The Originlab Software (Northampton, Massachusetts, USA) was used to analyze the XRD diffraction data and the Scherrer equation [26] applied to calculate the average size.

## Experimental animals

This was an in vivo experimental study, Wistar albino rats (8–10 weeks of age) weighing 150-220g, were acquired from the National Facility for Laboratory Animal Research and Care (NFLARC) at Panjwani Center for Molecular Medicine (PCMD), International Center of Chemical and Biological Sciences (ICCBS), University of Karachi. Prior to the experiments, the animals were acclimatized to the experimental environment for 14 days and allowed access to water and food ad-libitum. The animals were kept in standard propylene cages (25cmx40cmx50 cm; Height*Width*Length) with wood chips for bedding and nipple watering bottles, the animal beddings were changed three times a week. The animals were kept at 23-25°C with 12-hour light-dark cycles. Efforts were made to minimize animal suffering during the study period; on a daily basis, monitoring for any potential clinical signs of dehydration, weight loss, loss of appetite, pain, infection, abnormal behaviors and general physical appearance were made. Changes in body weight were monitored by taking a weekly record of the body weights were made to assess food in-take. Euthanasia criteria were set for experimental animals with significant health decline or unmanageable distress and this included persistent weight loss exceeding 20% of the initial body weight, behavioral changes such as severe lethargy, unresponsiveness, excessive grooming leading to self-injury, or vocalization, signal distress warranting euthanasia. Manifestation of physical signs, including extreme emaciation, dehydration, severe piloerection, or unhealed wounds and prolapses. Manifestation of neurological impairments such as seizures, tremors, or loss of coordination, as well as organ-specific toxicity like persistent diarrhea, or anuria. Similarly, unmanageable tumors exceeding 10% of body weight, ulcerated tumors, or systemic effects such as cachexia. Animals were monitored daily at 9:00am-10:00am and 2:00-5:00pm for any observable clinical signs or symptoms and behavioral changes. However, no rat met the euthanasia criteria, and no rat died before and during the experimental period. At the end of the study period, rats were anesthetized via subcutaneous injection of ketamine (50 mg/mL) and xylazine (0.2 mg/mL) at a ratio of 10:3 and a dose of 1 mL/kg body weight and sacrificed by cervical dislocation for the collection of blood and organs for the post-operative procedure. The body carcasses were placed in plastic bags, sealed off and stored in a deep freezer at -80°C, prior to incineration.

## Inclusion criteria

Female healthy Wistar albino rats of 8–10 weeks old and weighing between 150–220 g with no observable signs of illness or abnormalities and that had undergone a minimum 5–7 days of acclimatization to laboratory conditions before the experiment were included in the study

## Exclusion criteria

Animals exhibiting signs of illness, injury, or any pre-existing medical conditions. Pregnant or lactating female rats and those falling outside the specified weight range. Also, rats exhibiting signs of extreme stress or abnormal behavior during acclimatization and those previously used in other experiments were excluded from the study.

## Randomization

All animals that met the inclusion criteria were tagged with unique numerical numbers and the numbers for each animal were pooled together. The numbers were exported to Microsoft Excel software version 2016 (Microsoft office professional Plus 2016) and the *Rand() function* used to assign the animals to the treatment groups

## Blinding

The PFAgNPs and control substances (distilled water) were prepared in identical containers coded with alphabetical numbers. Only the study coordinator knew the actual content of each container. The technician who administered the test substances to the rats was unaware of the group assignments. Observations and recording clinical signs were done by another study member who was blinded of the treatment groups. Sample analysis such as hematological parameters, biochemical tests, histopathological analysis and data analyst were performed by members who were unaware of the treatment groups. The group identities were revealed only after all data had been collected, analyzed, and locked.

## Oral acute toxicity evaluation of PFAgNPs

Acute toxicity of PFAgNPs was conducted as following standard guidelines[27]. A total of fifteen (15) animals were randomly distributed into five treatment groups; Three (3) animals per group (5, 50, 300, 2000mg/kg PFAgNPs and Normal control (distilled water), respectively. The PFAgNPs were administered orally *via* gavage. After dosage, observations were made for acute signs of toxicity at 0.5h, 1h, 2h, 3h, 4h, 6h, 9h, 12h, and 24 h intervals and for the next 14 days where the animal survived. The injectable volume administered was calculated using the method below.

$$Volume\ of\ extract\ administered\ (mL) = \frac{Body\ weight\ (g)\ \times\ Dose\ (\frac{mg}{kg})}{concentration\ (\frac{mg}{mL})\ \times\ 1000}$$

The median lethal dose (LD50) was determined by identifying the lowest dose level that resulted in two or three mortalities. In this study, the lowest dose level that caused two mortalities out of three experimental animals was 2000 mg/kg. A retest at the same dose level yielded similar results. According to the criteria, the death of two animals at 2000 mg/kg estimated the $LD_{50}$ at 1000 mg/kg body weight. A total of 18 animals were used in this study and only 4 died at a dose of 2000mg/kg PFAgNPs.

## Induction of uterine fibroids and treatment

Thirty (30) female virgin Wistar rats were randomly assigned to six groups, each comprising five animals. Uterine fibroids were induced in all groups (I-V) by a daily single oral administration of 600 mg/kg MSG with distilled water as the vehicle for 14 days. After induction, the rats received treatment with PFAgNPs for 14 days: 25 mg/kg, 50 mg/kg, and 100 mg/kg of PFAgNPs, Standard drug control 10 mg/kg Ulipristal acetate, Positive/disease control: 600 mg/kg MSG without any treatment, Normal control; 1mL/100 g body weight/day of distilled water only. The treatment doses were selected based on preliminary experiments. After the experimentation period, the rats were euthanized under ketamine-xylazine anesthesia (100 mg/kg BW, intraperitoneally). Blood and tissue were collected for biochemical analysis.

## Biochemical analysis

The total serum estrogen, protein, triglycerides, progesterone and cholesterol were determined by collecting whole blood by cardiac puncture into non-heparinized tubes. Then blood was centrifuged at 3000rpm for a total of 20 minutes and Randox kits used for the analysis following the manufacturer's instructions. Uterus samples were also collected and preserved in 10% neutral for H and E histological staining and analysis.

### Gross pathological observations and histopathological studies

The uterine tissues were harvested, weighed, and visually inspected for any histomorphological changes and preserved in 10% neutral buffered formalin. Tissue processing was performed followed by Hematoxylin and Eosin (H&E) staining according to a standard protocol [28]. Tissue samples were visualized on a Nikon Ni-E microscope, and images were captured using NIS element advanced research software at the national imaging facility at PCMS, ICCBS, University of Karachi. The tissue samples were analyzed by a histopathologist who was blinded to the experiments.

### Statistical analysis

All numerical data collected was processed using Microsoft Excel spreadsheet then exported to Graph Pad Prism software version 5.0 (San Diego, USA) and analyzed using one-way analysis of variance (ANOVA). Statistical significance was determined at $P$ value $< 0.05$.

### Compliance with ethical standards

The study acquired ethical clearance (KNH-ERC/RR/761) from Kenyatta National Hospital/University of Nairobi-Ethics and Research Committee (KNH/UON-ERC) and to conduct animal experiments; approval (ICCBS-ASP-02-2024-05) was obtained from the ethical committee of animal research, Dr. Panjwani Center for Molecular Medicine and Drug Research (PCMD), International Center for Chemical and Biological Sciences (ICCBS), University of Karachi. All experiments were conducted in accordance with the ARRIVE guidelines(29). Clinical trial number not applicable for this study.

## Results

### Characterization of the synthesized PFAgNPs

The maximal absorbance at a wavelength of approximately 425 nm (**Fig.1A**), confirmed the successful formation of PFAgNPs. At various temperatures, i.e., 25-85°C and pH (2–11), the synthesized PFAgNPs were found stable as they maintained the maximum absorbance within the range of 430–445 nm (Fig 1B -1D). The PFAgNPs demonstrated an average hydrodynamic size distribution of 107.4 d. nm, a polydispersity index of 0.264 and a zeta potential of -18.3Mv (**Fig 2**). The PFAgNPs FTIR spectrum revealed a shift in the peaks to a lower wavenumber and %transmittance as compared to those of the PFE spectrum; confirming the involvement of PFE phytocompounds in the reduction of silver ions and eventual synthesis of the PFAgNPs. The FTIR spectrum in **Fig 3** shows important peaks at 826.26, 1030.27, 1258.8, 1352.7, 1713.95, 2928.52 and 3272.32 cm$^{-1}$. The peak at 3272.32 cm$^{-1}$ corresponds to O-H stretching, whereas the one at 2928.52 cm$^{-1}$ is attributed to the C-H stretch for sp3-hybridized carbon atoms, highlighting the role of alcohols and alkanes as the capping agent for AgNPs for the stability of PFAgNPs respectively. The bands at 1713.96 and 1258.88 can be attributed to the carbonyl C=0 stretch, while the bands at 1352.7 cm$^{-1}$ and 1603.05 cm$^{-1}$ respectively correspond to the –C=C– and –C-O stretch vibration of antioxidants (**Fig 3**). The four diffraction peaks(38.48°, 44.33°, 64.28°, and 75.399°) recorded in XRD analysis correspond to lattice plane values of the face centered cubic(FCC) silver, i.e., (111, 200, 220 and 311) with a lattice parameter of a = 4.12 Å which were are in consonance with structure of silver from joint committee of powder diffraction standard (JCPDS) Card No-087–0720 (**Fig 4**). The AFM analysis revealed the spherical shape of the PFAgNPs (**Fig 5A**). The 3D images of PFAgNPs indicated an average NP height of 25 nm (**Fig 5B**). Silver had the highest concentration in the PFAgNPs as per the EDX spectra (**Fig 6B**). The SEM micrograph revealed the spherical nature of the synthesized PFAgNPs (**Fig 6C** and **Fig 6D**).

### Oral acute toxicity evaluation

At 5 and 50 mg/kg, the animals exhibited normal behavior with no major signs of toxicity observed (**Table 1**). At 300 mg/kg, the animals exhibited mild forms of toxicity such as increased activity, lethargy, piloerection and hyperventilation. These effects lasted for 10–15 minutes post-administration, and full recovery was recorded. No mortality was observed even

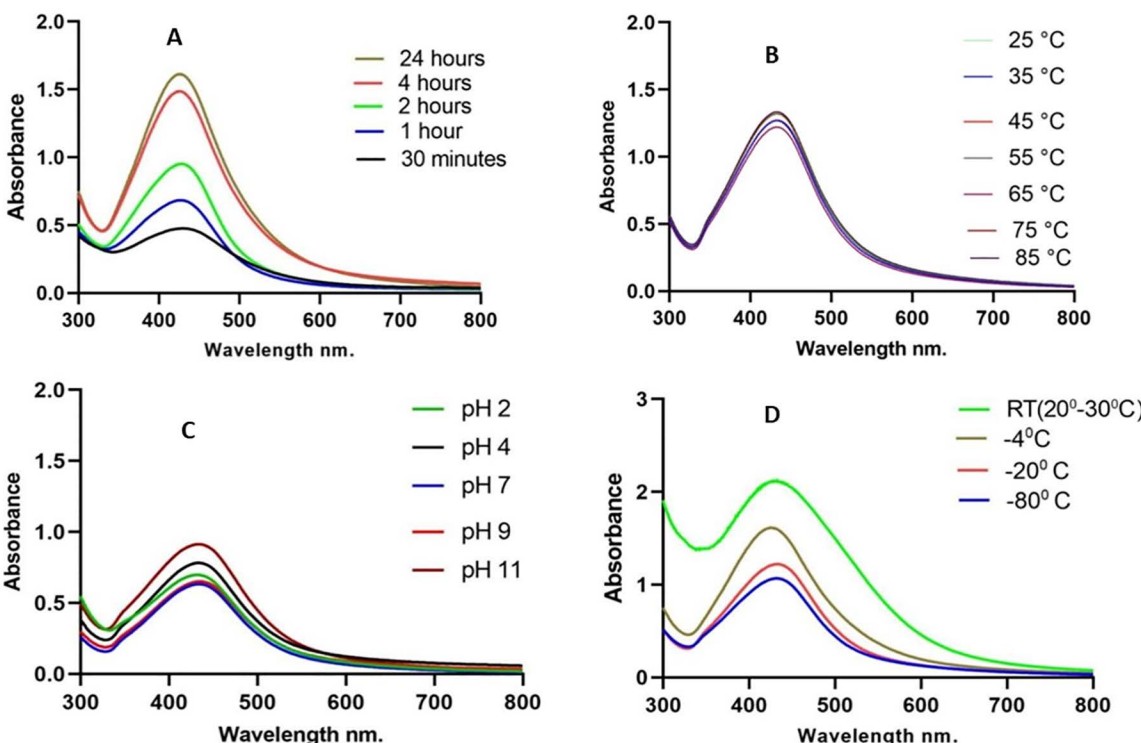

**Fig 1.** UV/VIS spectra of PFAgNPs; (A) After 24 hours of incubation, (B) Showing temperature stability, (C) Showing pH stability (D) Showing storage stability after 6 months under different storage conditions.

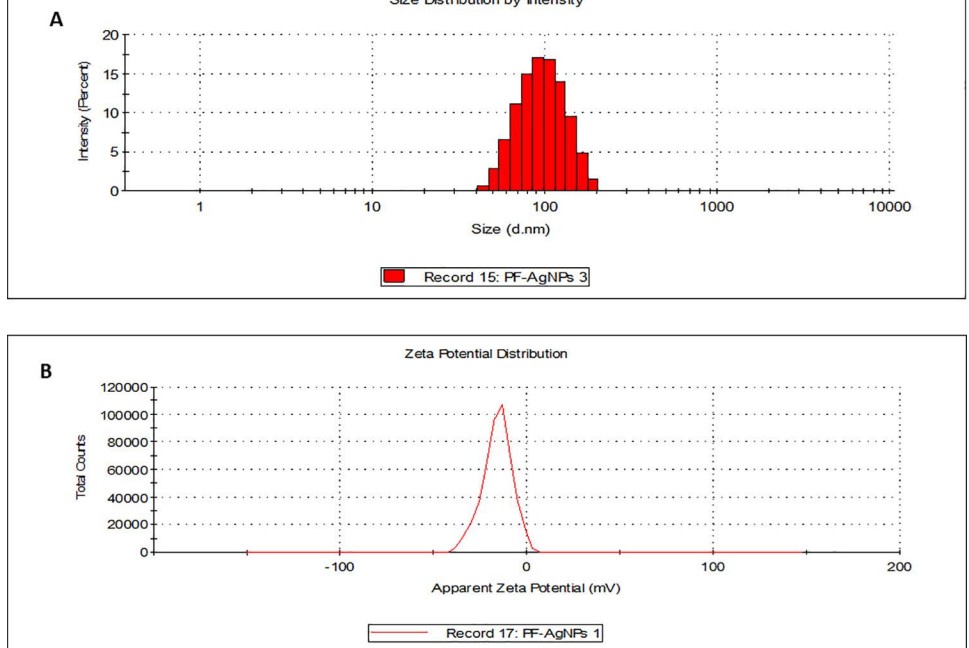

**Fig 2.** (A) Size distribution by Intensity, (B) Zeta potential distribution of the synthesized PFAgNPs.

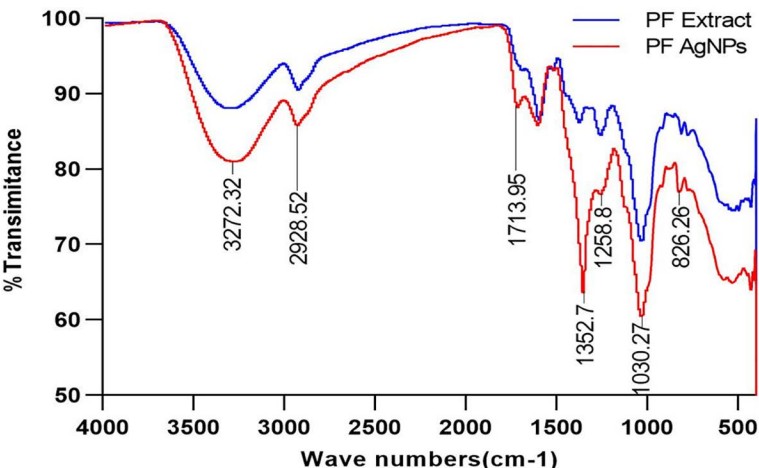

**Fig 3. FTIR spectra of the synthesized PFAgNPs and Polyscias fulva aqueous extract.**

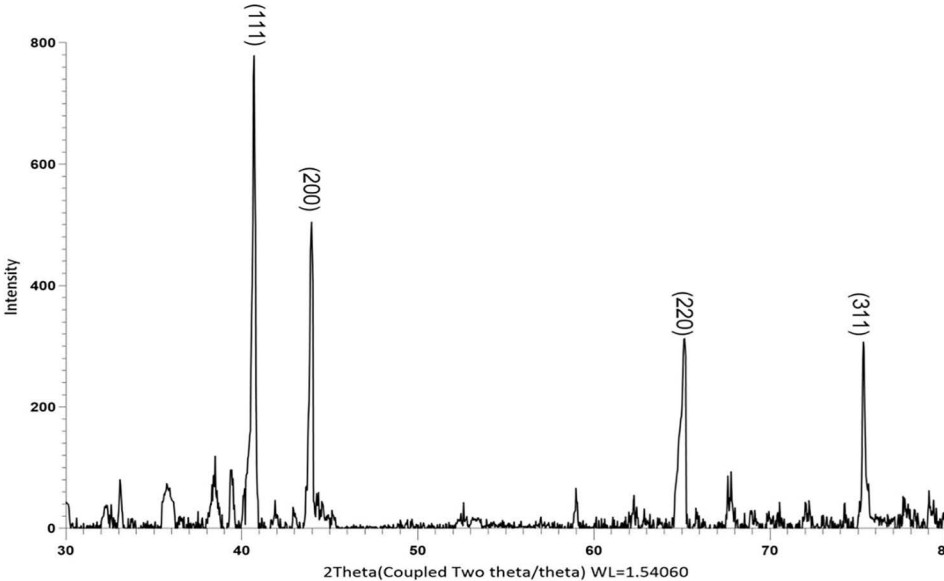

**Fig 4. XRD diffraction pattern spectra of PFAgNPs.**

after 14 days. At 2000mg/kg, severe toxicity behaviors were observed characterized by the above-mentioned effects with additional loss of a righting reflex, unusual vocalization, hyperventilation, writhing, convulsions and ultimately death of two animals. The median lethal dose ($LD_{50}$) of the PFAgNPs was determined to be 1000 mg/kg.

### Effects of PFAgNPs on uteri weights, reproductive hormones, and biochemical parameters in MSG-treated female rats

The variations in uterine weight coefficients, biochemical indices, and hormonal factors between the disease control and normal control groups, demonstrate that MSG significantly induced uterine fibroids in the Wistar rats ($P < 0.05$). The

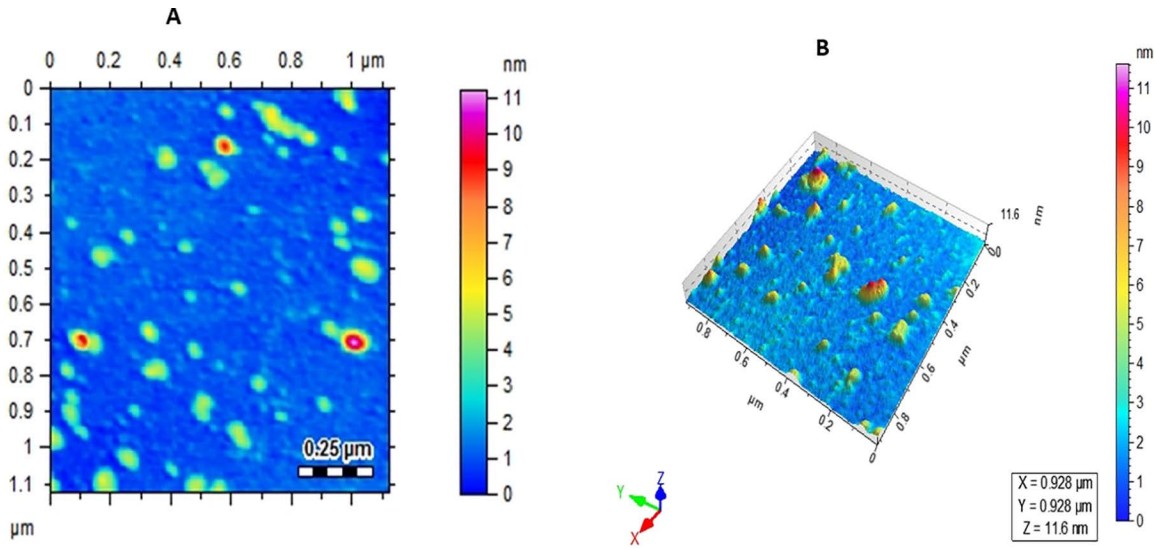

**Fig 5. Atomic Force Microscopic images of the synthesized PFAgNPs, 2D (A) and 3D (B).**

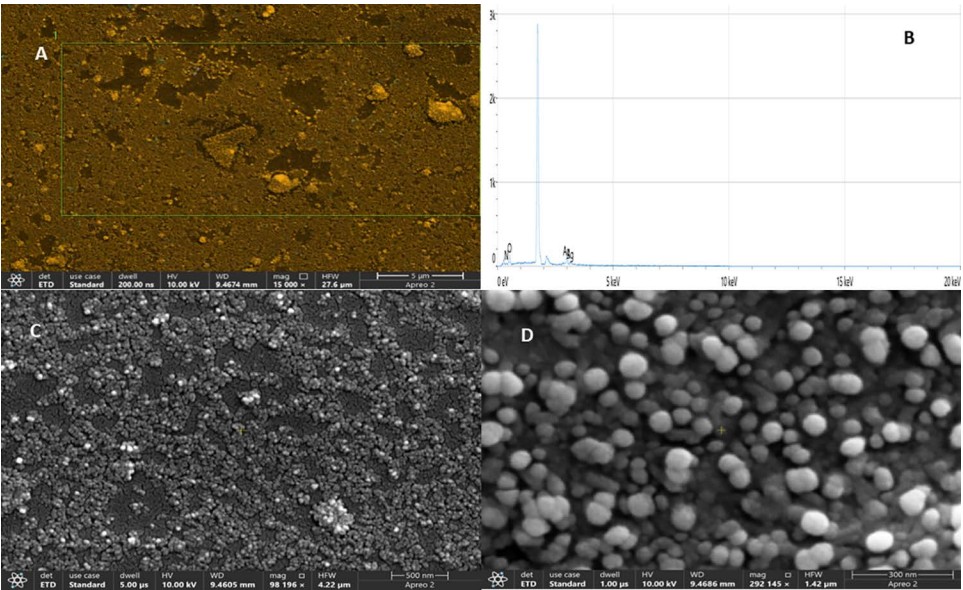

**Fig 6. (A) Mapped region for EDX Spectra (B) The quantitative amounts of different elements present in the PFAgNPs with Ag showing the highest concentration in the EDX spectra.** (C) And (D) SEM micrograph at different resolution showing the shape of the Synthesized PFAgNPs.

MSG-treated animals (disease control group) had significantly high (P<0.05) uteri weights when compared to the normal controls (**Fig 7**). Additionally, significantly high serum levels of estrogen (P=0.0001), progesterone (P=0.0003), cholesterol (P=0.0001), globulin (P=0.0001), albumin (P=0.05), total protein (P=0.0001), and were also noted compared to the normal control. The PFAgNPs treated groups had significantly low uteri weights (P<0.05) when compared to the disease control group. Furthermore, the serum estrogen levels were significantly lower (P<0.0001) after 25 mg/kg PFAgNPs than

**Table 1. Table showing the acute toxicity effects of PFAgNPs on the general behavior of the Wistar albino rats.**

| Acute Effects | GROUP | | | |
|---|---|---|---|---|
| | 5 mg/kg | 50 mg/kg | 300 mg/kg | 2000 mg/kg |
| Decreased activity | 0/3 | 0/3 | 0/3 | 0/3 |
| Increased activity | 1/3 | 1/3 | 2/3 | 3/3 |
| Salivation | 0/3 | 0/3 | 0/3 | 0/3 |
| Lacrimation | 0/3 | 0/3 | 0/3 | 0/3 |
| Diarrhea | 0/3 | 0/3 | 0/3 | 0/3 |
| Lethargy | 0/3 | 2/3 | 3/3 | 3/3 |
| Pilo erection | 0/3 | 2/3 | 3/3 | 3/3 |
| Loss of a righting reflex | 0/3 | 0/3 | 0/3 | 3/3 |
| Unusual vocalization | 0/3 | 0/3 | 2/3 | 3/3 |
| Hyperventilation | 0/3 | 1/3 | 2/3 | 2/3 |
| Writhing | 0/3 | 0/3 | 1/3 | 3/3 |
| Convulsions | 0/3 | 0/3 | 1/3 | 2/3 |
| Death | 0/3 | 0/3 | 0/3 | 2/3 |

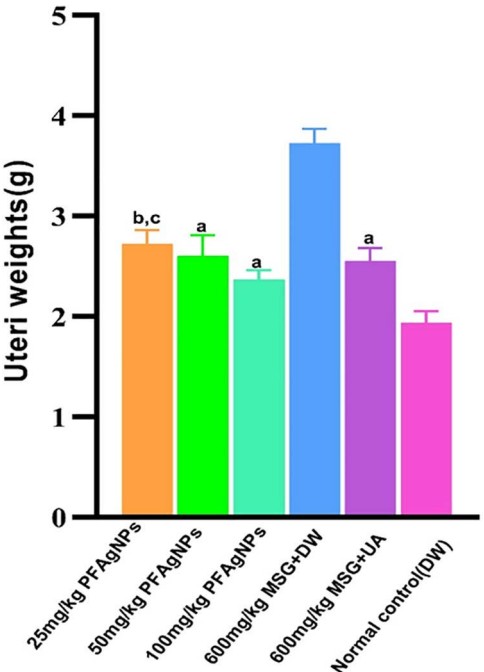

**Fig 7. Effects of PFAgNPs on Uteri weight coefficients in MSG-Uterine fibroids in Female Wistar rats.**

in the disease control group (**Fig 8A**). However, compared to the normal control group, the levels were significantly high (P < 0.0028). Higher doses (50 mg/kg and 100 mg/kg) were associated with significantly low estrogen levels (P < 0.0001) in comparison to the disease control.

In regard to serum Progesterone, there were no statistically significant differences in the concentration between 25 mg/kg, 50 mg/kg PFAgNPs and the disease control. However, the 100 mg/kg PFAgNPs treated group had significantly

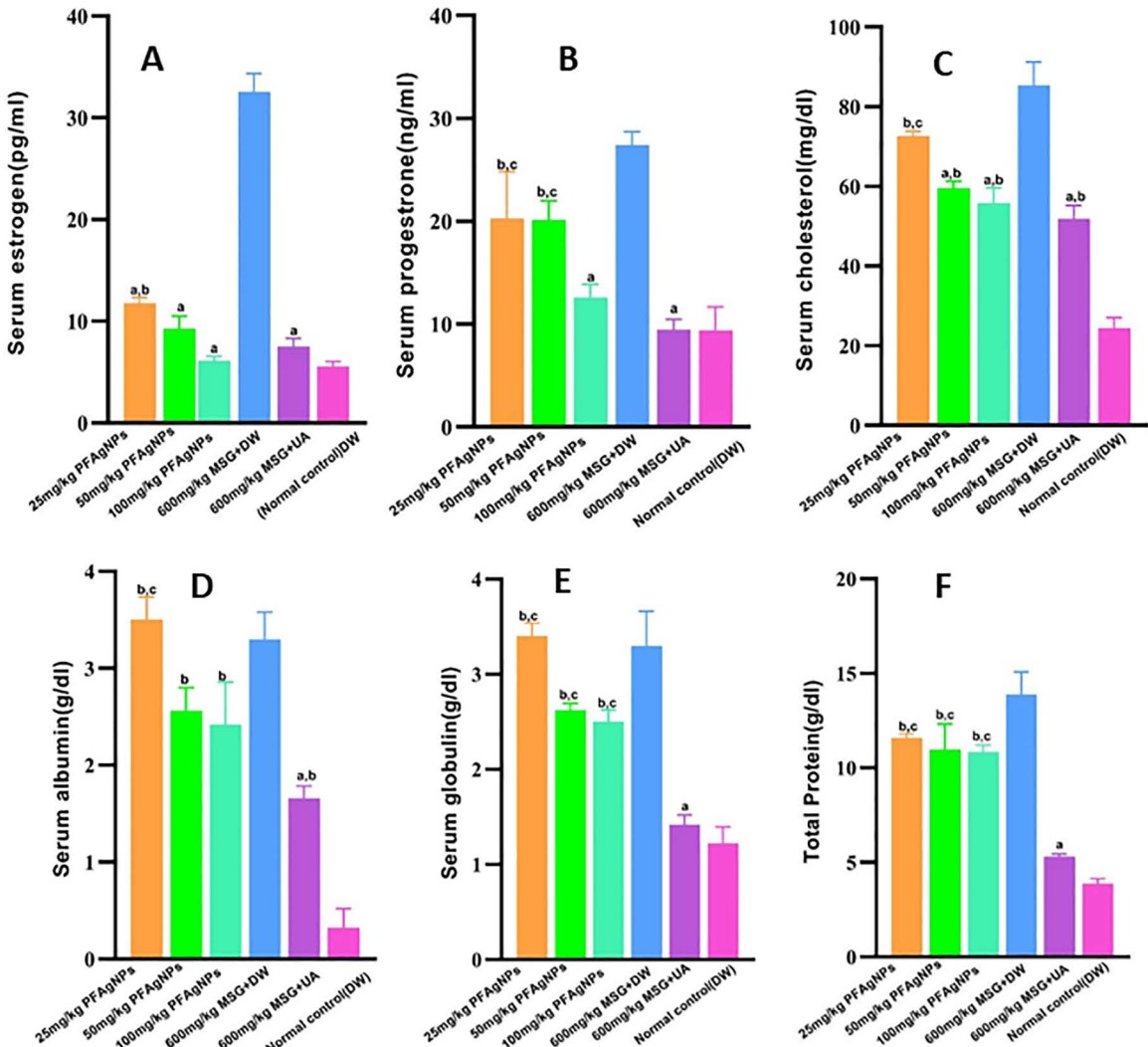

**Fig 8. Effect of PFAgNPs on serum (A) estrogen, (B); Progestrone, (C); cholesterol, (D) Total protein, (E) Albumin and (F) globulin against MSG induced uterine fibroids in female Wistar rats, a:P-value<0.05 as compared to disease control(600mg/kg MSG+DW), b: P-value<0.05 as compared to the Normal control(DW only), c:P-value<0.05 as compared to the standard treatment control(600mg/kg MSG+10mg/kg UA), MSG: Monosodium glutamate, DW: Distilled water, UA: Upristal acetate.**

($P$=0.0025) low serum progesterone levels in comparison to the disease control (**Fig 8B**). No significant differences in blood cholesterol for the 25mg/kg PFAgNPs group in comparison to the disease control. However, animals that received 50mg/kg (P=0.0003) and 100mg/kg (P<0.0001), had significantly low serum cholesterol levels as compared to disease control (**Fig 8C**). Notably, there was no discernible difference in serum cholesterol levels between the standard treatment group (10mg/kg Ulipristal acetate) and 50mg/kg and 100mg/kg groups. Serum levels of albumin, globulin, and total protein did not significantly differ across treatment groups when compared to the disease control (**Fig 8D**–**8F**).

## Effects of the PFAgNPs on uterine histology in MSG-Induced uterine fibroids in Wistar Rats

Oral administration of MSG at 600mg/kg caused notable cellular lesions in the Hematoxylin and Eosin-stained uterine sections; severe inflammation in the endometrium associated with degeneration and cellular apoptosis was observed;

notably, the stromal cells of the endometrium were markedly disorganized and revealed vacuolar degeneration. Marked disorganization, hyperplasia and loss of the columnar shape of the epithelial cells were also noted. Treatment with PFAgNPs at 100mg/kg caused a moderate degree of inflammation with apoptosis; moderate degeneration of the endometrium layer was noted alongside some degree of epithelial cell disorganization and congestion of the blood vessels. Interestingly, lower doses (25 mg/kg and 50 mg/kg) of PFAgNPs ameliorated the MSG-induced uterine tissue necrosis and apoptosis, revealing mild inflammation with minor necrosis on the endometrium layers with no major damage on the myometrium and perimetrium; The PFAgNPs at these doses preserved the columnar shape on the epithelial cells and the blood vascularity of the glands was within normal histological limits. The normal control group exhibited a normal histoarchitecture with no significant cellular lesions in the endometrium, myometrium and perimetrium; no necrosis nor apoptosis was noted in the uterine tissues (**Fig 9**).

## Discussion

*Polyscias fulva* is a traditional plant used for the management of uterine fibroids and cancer in East and Central Africa [8,29]. The plant's phytocompounds, such as alpha-hederin, quercetin, β-sitosterol [10], contribute to it's pharmacological properties [29]. These compounds, exhibit several pharmacological properties such as inhibiting abnormal cell growth, making *Polyscias fulva* a promising candidate for fibroid and tumor therapy [8].

The green synthesis method, is a not only a cost effective but also an eco-friendly approach to enhance the therapeutic efficacy of AgNPs [22]. Plants provide a suitable alternative for the synthesis of NPs due to their strong reducing power attributed to the rich phytochemicals. The color change observed during the synthesis of our PFAgNPs is attributed to the oscillation of conduction band electrons of Ag as a result of exposure to the reducing phytocompounds in *Polyscias fulva*

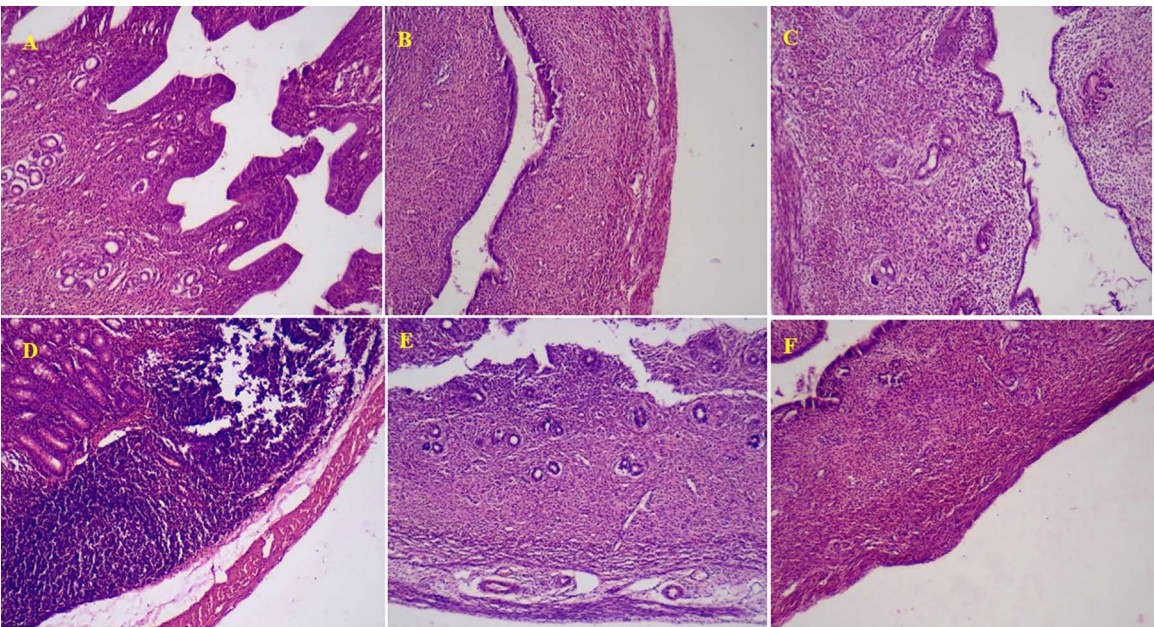

**Fig 9. Photomicrographs of Uterine tissue sections stained by Hematoxylin and Eosin in MSG-treated female rats (X 400).** A; 25mg/kg PFAg-NPs, B: 50 mg/kg PFAgNPs; Mild chronic inflammation noted in the endometrial epithelial cells alongside inflammatory cell infiltration; C; 100mg/kg PFAgNPs: Marked inflammation in the endometrial epithelial cells D; 600 mg/kg MSG: Severe chronic inflammation noted in the endometrial epithelial cells alongside vacuolar degeneration and inflammatory cell infiltration, E: 600mg/kg MSG+ Ulipristal acetate; mild chronic inflammation noted in the endometrial epithelial cells. F: Normal control; No hyperplasia, atrophy, edema, denaturation or necrosis noted in the endometrial epithelial cells..

extract, a phenomenon called the surface plasmon resonance (SPR) [30]. The UV–visible spectrophotometer maximal absorbance of the PFAgNPs at a wavelength of 427 nm falls within the absorption range (approximately 400–450nm) of the surface plasmon resonance for AgNPs as reported from previous studies[21,23]. This was achieved within 24 hours at room temperature and at a dilution of 1:50 (*Polyscias fulva* extract: silver nitrate solution) without any additional reducing or stabilizing agents. This demonstrates the potential of synthesizing a high quantity of PFAgNPs while utilizing minimal natural resources, offering an opportunity for sustainable use and conservation of plants.

The stability of AgNPs under different physical conditions is crucial for assessing their nature and retention of biological activity[31]. The PFAgNPs remained stable at the different tested temperature conditions maintaining an absorbance range of 420–450 nm, characteristic of AgNPs. This remarkable property could imply that even when subjected to varying storage or body temperatures conditions, the PFAgNPs can maintain their nature and behavior without losing their activities, making them suitable for several biological and medical applications [32].

The pH conditions significantly influence the stability of AgNPs by affecting their aggregation, surface charge, chemical reactivity, and interactions with surrounding molecules. Understanding the pH-dependent stability of AgNPs is therefore essential for uses such as drug delivery, where physiological pH levels may affect the behavior and effectiveness of the nanoparticles. The synthesized PFAgNPs were stable when exposed to varying pH 2–11 conditions with absorbance peaks ranging from 415–430nm typical of AgNPs [31]. Extreme pH conditions can induce AgNPs aggregation or precipitation [32]. For instance, at very low or high pH values, aggregation of the nanoparticles may occur due to reduced electrostatic repulsion or changes in surface chemistry [33]. This aggregation can compromise stability and alter the physicochemical properties of AgNPs [34]. The demonstrated pH stability of the PFAgNPs implies that they could as well remain stable when exposed to the acidic pH media of the stomach or the basic pH media of the ileum if used *in vivo* [25]. This property is not common with plant crude extracts that often face absorption challenges due to the varying gastrointestinal tract pH affecting their bioavailability and, ultimately, their efficacy [35]. Furthermore, the PFAgNPs remained stable for 6 months at different storage conditions with maximal absorption peaks ranging between 417–430nm. In particular, PFAgNPs preparations kept at room temperature increased in absorbance as compared to the other PFAgNPs kept at other storage conditions. This agrees with other studies and attributed to the continuous formation of AgNPs as a result of exposure to the same synthetic conditions [25].

The size, shape, surface area, and polydispersity index characterize the AgNPs. The average hydrodynamic size falls within the acceptable ranges of AgNPs, and the low PDI indicates the high homogeneity and good quality of the PFAgNPs, inferring the possibility of their effectiveness as nano-drug carriers [36]. The negative zeta potential demonstrated by the PFAgNPs could be attributed to the hydroxyl groups of polyphenols present in PFE [11], exerting strong electrostatic repulsion between particles. This behavior helps prevent aggregation and maintain the nanoparticles' stability [37]. An aggregate of all these features indicates the successful synthesis of highly stable nano-sized AgNPs with the potential for several applications.

The AFM, SEM, XRD and FTIR findings align with the expected characteristics of AgNPs in terms of shape and structure, as reported in previous studies [31]. The DLS analysis estimated the average hydrodynamic size of the PFAgNPs as 107.4 d.nm, which is way higher as compared to the SEM (35.1 nm) and XRD (25.28nm) reports. This divergence could be attributed to the fundamental principles of each technique and the conditions under which the measurements were performed. DLS measures the hydrodynamic diameter, which includes not only the nanoparticle core but also the surrounding solvation layer, adsorbed biomolecules, and any potential agglomeration in solution[22,25,38]. As a result, the size determined by DLS is typically larger than that obtained from other methods. In contrast, XRD provides an estimate of the crystalline domain size based on peak broadening, which does not account for particle aggregation or surface coatings, leading to a smaller calculated size[39]. Whereas, SEM captures direct images of dry nanoparticles, providing an intermediate size estimation, as particles may undergo some degree of aggregation or shrinkage due to sample preparation conditions[40]. The observed discrepancies highlight the importance of employing multiple complementary techniques to accurately characterize nanoparticle size and state. Furthermore, the potential for size variation under biological conditions due to aggregation, protein corona formation, or dissolution remains an important consideration. Future studies

should focus on assessing the colloidal stability of PFAgNPs in biological media to better understand their behavior under physiological conditions.

*Polyscias fulva* stem bark has been reported to possess several phytochemical compounds including flavonoids, terpenoids and alkaloids. The hydroxyl and carbonyl functional groups recorded by the FTIR analysis, could be attributed to the above phytocompounds which are powerful reducing agents and can reduce $Ag^+$ ions to $Ag^0$ resulting into nanoparticle synthesis. The alcohol, carbonyl and antioxidant functional groups could still be attributed to the flavonoids and phenolics which possess a strong affinity to bind metal ions and may be encapsulated around the nanoparticles, forming a protective coat-like membrane preventing the agglomeration and thus resulting in nanoparticle stabilization in the medium [21]. These findings further confirmed that phytoconstituents contribute as reducing and stabilizing agents [36].

The acute toxicity assessment of PFAgNPs revealed dose-dependent adverse effects, including lethargy, hyperventilation, and mortality at higher dosages. One possible explanation for these toxic manifestations is nanoparticle-induced oxidative stress at high doses, a well-documented phenomenon in nanoparticle toxicity[41]. AgNPs have been reported to generate reactive oxygen species (ROS) at high doses, leading to oxidative damage to cellular components, lipid peroxidation, and mitochondrial dysfunction[41]. This could trigger systemic toxicity, manifesting as physiological distress symptoms such as lethargy and respiratory irregularities. Additionally, nanoparticle exposure may elicit an inflammatory response AgNPs have been reported to activate immune pathways, leading to the release of pro-inflammatory cytokines such as interleukin-6 (IL-6), tumor necrosis factor-alpha (TNF-α), and interleukin-1 beta (IL-1β)(43). Excessive inflammation could contribute to respiratory distress and systemic toxicity, potentially explaining the observed hyperventilation and mortality at higher doses.

Although this study did not directly measure oxidative stress markers (e.g., malondialdehyde levels, glutathione depletion) or inflammatory cytokine responses, we recognize their importance in providing mechanistic insights into PFAgNP-induced toxicity. Future studies should focus on evaluating these biochemical markers to better understand the pathways involved and assess potential mitigation strategies. Furthermore, the observed acute toxicity effects could possibly be indicative of the neurological effects of AgNPs at high doses, whose minute size favors them to cross the blood-brain barrier into the Central Nervous system, where they interact with various neuronal cells, disrupting neurotransmitter signaling and neuronal function[42]. The $LD_{50}$ of PFAgNPs was calculated as 1000 mg/kg, and according to the globally harmonized classification system, the PFAgNPs fall within category 4 of the toxicity class, i.e., moderately toxic [27].

Uterine fibroids (UF) are the most common type of gynecological disease. While often asymptomatic and lacking distinct symptomatology, their pathogenesis and progression are strongly associated with significant alterations in serum proteins, cholesterol, and female reproductive hormones [39]. Changes in uterine weight provide valuable insight on the effects of test compounds on the uterus. In this study, the PFAgNPs mitigated the MSG-induced increase in uterine weight. In contrast, copper nanoparticles significantly reduced the relative uterine weight coefficients in rats after 28 days of treatment. Similarly, molybdenum metal oxide nanoparticles ($MoO_3$NPs) also caused a notable decrease in uterine weight.

The significant decrease in total proteins, albumin and globulin in the PFAgNPs treated rats as compared to the disease control infers the potential of the PFAgNPs to control the MSG-associated elevation of the above proteins. AgNPs interaction with serum proteins and alterations of their concentrations have also been reported in other studies [39–41].

Elevated serum lipid profile levels have been strongly associated with the pathogenesis of uterine fibroids [43]. The significant decrease in serum cholesterol indicates the ability of the PFAgNPs to regulate the regulate cholesterol homeostasis. The findings are in line with other studies where AgNPs-exposure to Wistar rats decreased serum high-density lipoprotein cholesterol and, on the contrary, elevated the total cholesterol, triglycerides, low-density lipoprotein cholesterol and glycerol [43]. The hypocholesterolemia effect of the AgNPs is attributed to their antioxidant properties at lower doses, which help mitigate oxidative stress-mediated lipid peroxidation and reduce the generation of reactive oxygen species [44]. By controlling oxidative damage, AgNPs may play a role in regulating cholesterol levels and maintaining lipid homeostasis. The

significant decrease in serum estrogen and progesterone noted in the PFAgNPs treated group compared to the disease control group, align with other studies showing that administering AgNPs to mice lowers serum estrogen levels, disrupting endocrine function and altering mammary gland development[45,46]. MoO$_3$NPs significantly decreased serum estrogen levels in female rats[47], whereas the Titanium-dioxide (TiO$_2$) nanoparticles dose-dependently decreased serum Follicle stimulating and Luteinizing hormone levels while lowering Estrogen levels in female rats[48]. Furthermore, significant reductions in estrogen and progesterone were caused by aluminum oxide nanoparticles [49]. Due to their nano-sizes, the NPs to can cross the blood-brain barrier, potentially affecting the hypothalamic-pituitary pathway hence causing these effects[48]. This interference can disrupt the balance of the critical neurohormones which play a key role in the regulation of ovarian function through positive and negative feedback mechanisms[50]. The AgNPs have also been reported to directly affect the ovaries through follicle function interference hence inhibiting the growth of the estrogen-secreting follies and ultimately decreasing estrogen serum concentrations [48]. The AgNPs also decrease the Estrogen receptor α expression, reduce the epithelial proliferation, and diminish the fibrous collagen deposition around the epithelium [51]. Estrogen and progesterone are known to play key roles in the growth and development of uterine fibroids by regulating cellular proliferation, extracellular matrix production, and angiogenesis[3]. Estrogen, in particular, has been shown to stimulate fibroid growth through its receptor-mediated signaling, while progesterone's role remains more complex, with both proliferative and inhibitory effects[52]. The modulation of these hormones by PFAgNPs could potentially alter their balance, thus contributing to fibroid regression. Additionally, serum parameters such as cholesterol and protein levels may reflect systemic alterations induced by PFAgNPs[3,48]. Cholesterol, which is integral to cell membrane integrity and steroid hormone synthesis, could influence fibroid pathophysiology by affecting cellular signaling and metabolic pathways[48,53]. Proteins, including various enzymes and cytokines, might also play a role in mediating the anti-inflammatory and anti-proliferative effects of PFAgNPs, contributing to fibroid regression[54].

The histomorphological and histopathological assessments of the uterine tissue support our serum biochemical and hormonal assay results. The MSG-only treated animals experienced severe chronic inflammation, hyperplasia, and loss of columnar shape, whereas the PFAgNPs treated groups exhibited less damage. MSG has been reported to affect the reproductive enzymes, particularly estrogen, and uterine histoarchitecture due to its ability to activate the enzyme aromatase [55–57] which plays a catalytic role in the conversion of testosterone to estradiol [58]. Elevated estradiol levels can directly induce changes in uterine tissue, contributing to the observed alterations in reproductive enzyme activity and uterine histology[55].

Strikingly, lower doses (25 mg/kg, 50 mg/kg) of PFAgNPs exhibited greater protection against the MSG-induced effects than the higher dose (100 mg/kg), resulting in mild inflammation. AgNPs and other metal nanoparticles (Cu, ZnO, TiO$_2$) have been associated with reproductive toxicity, including damage to reproductive and somatic cells, ovarian failure, and follicular abnormalities[48,59,60]. The high dose AgNPs have been reported to cause marked histological alterations of the uterine tissue in experimental mice after sub-chronic intraperitoneal administration [46]. Furthermore, AgNPs significantly affected the serum concentrations in the sex hormones, histoarchitecture of the reproductive organs, and induced oxidative stress resulting into cellular apoptosis. Therefore, the tissue damage exhibited by the high dose (100 mg/kg) PFAgNPs group could be attributed to the innate uterine toxicity of the AgNPs [61].

However, there is a need for a more detailed morphometric analysis to quantify tissue changes in terms of cell size, density, and fibroid volume, which would offer a more precise assessment of the extent of fibroid regression or progression following treatment with PFAgNPs. Additionally, implementing uterine tissue grading score could provide a standardized method for evaluating the severity of histopathological changes, allowing for a more detailed comparison of treatment effects across different dosages or time points. These methods would strengthen the histological analysis and improve the interpretation of therapeutic outcomes.

This study's main limitation is the lack of a comprehensive dose-response analysis for PFAgNPs, which prevents precise determination of the optimal therapeutic dose that maximizes efficacy while minimizing adverse effects. Further

research should evaluate a broader range of doses with histopathological assessments to refine the clinical application of PFAgNPs.

Additionally, future studies should integrate Waveflex biosensors for real-time monitoring of biochemical and hormonal changes, providing high-resolution data on estrogen, progesterone, cholesterol, and inflammatory markers. W-shaped optical fiber (FWOF) biosensor coupled with gold nanoparticles (AuNPs) have been developed to detect serum histamine[62], similarly, ZnO-Nanowires and tungsten disulfide quantum dots Functionalized Multicore Fiber-Based W-Shaped Waveflex Biosensors have also been developed for Rapid Detection of Hemoglobin A1c [63] alongside several other biosensors designed for cancer biomarker detection[64]. Leveraging these advanced biosensing technologies could enable to obtain high-resolution, continuous data on estrogen and progesterone levels, allowing for a more precise assessment of how PFAgNPs modulate hormonal balance in fibroid development and regression. Additionally, Waveflex biosensors could be utilized to track systemic biomarkers such as cholesterol, inflammatory cytokines, and extracellular matrix-related proteins. This could offer deeper insights into the metabolic and molecular mechanisms through which PFAgNPs exert their anti-fibrotic effects

Further investigations should also incorporate in vitro human cell studies using uterine or fibroblast cell lines to assess toxicity, bioavailability, and interactions with hormone receptors. Advanced models such as 3D cultures or organ-on-a-chip systems could better simulate human physiology, aiding in optimizing dosing strategies before clinical trials.

## Conclusion

In conclusion, the PFAgNPs exhibited an ameliorative effect against monosodium glutamate-induced uterine fibroids in female Wistar rats. Lower doses effectively reduced serum proteins, cholesterol, and female reproductive hormones while preserving the histoarchitecture of the uterine tissues. However, despite having similar effects on serum assays, higher doses are non-protective against MSG-induced histopathological effects on the uterine tissues. Acute toxicity studies indicated that while lower doses exhibited minimal adverse effects, higher doses led to severe toxicity and mortality in Wistar albino rats. In this study, in vivo experimental approaches were used to determine the safety and efficacy of the PFAgNPs in the management of UF, However the physiology of the Wistar rats differs from that of humans, hence this limits the direct extrapolation of findings to human patients. Despite the promising results, further advanced mechanistic studies and clinical trials are required to further assess the therapeutic benefits and safety of PFAgNPs in treating uterine fibroids and their potential and safe application in various fields.

## Acknowledgments

The authors gratefully acknowledge the support provided by Mr. Aziz Muwaba, and the local leadership of Najjembe village, Buikwe district, for their assistance during the field collection of plant materials. We also extend our appreciation to Ms. Edeya Sharon and Mr. Emmanuel Omondi, the Laboratory technicians at the Natural Products Research and Innovation Center, Busitema University for their valuable support in the post-harvest processing and extraction of plant materials in the laboratory.

## Author contributions

**Conceptualization:** Kenedy Kiyimba, Samuel Baker Obakiro, Were Lincoln Munyendo, Eric M. Guantai, Yahaya Gavamukulya.

**Data curation:** Kenedy Kiyimba, Were Lincoln Munyendo, Eric M. Guantai, Yahaya Gavamukulya.

**Formal analysis:** Abdul Jabbar, Samuel Baker Obakiro, Eric M. Guantai.

**Funding acquisition:** Ayaz Ahmed, Mohammad Iqbal Choudhary, Were Lincoln Munyendo.

**Investigation:** Kenedy Kiyimba, Khadija Rehman, Syed Mehmood Hasan, Abdul Jabbar, Samuel Baker Obakiro, Eric M. Guantai, Yahaya Gavamukulya.

**Methodology:** Kenedy Kiyimba, Khadija Rehman, Syed Mehmood Hasan, Abdul Jabbar, Muhammad Raza Shah, Yahaya Gavamukulya.

**Project administration:** Mohammad Iqbal Choudhary.

**Resources:** Ayaz Ahmed, Mohammad Iqbal Choudhary, Muhammad Raza Shah, Were Lincoln Munyendo, Yahaya Gavamukulya.

**Supervision:** Ayaz Ahmed, Samuel Baker Obakiro, Muhammad Raza Shah, Were Lincoln Munyendo, Eric M. Guantai.

**Validation:** Kenedy Kiyimba.

**Visualization:** Muhammad Raza Shah.

**Writing – original draft:** Kenedy Kiyimba.

**Writing – review & editing:** Kenedy Kiyimba, Ayaz Ahmed, Mohammad Iqbal Choudhary, Khadija Rehman, Syed Mehmood Hasan, Abdul Jabbar, Samuel Baker Obakiro, Muhammad Raza Shah, Were Lincoln Munyendo, Eric M. Guantai, Yahaya Gavamukulya.

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
