## [Decision Letter · Decision Letter 0]

7 Mar 2025

PONE-D-24-56554Green synthesized Polyscias fulva silver nanoparticles ameliorate Uterine Fibroids in Female Wistar Albino RatsPLOS ONE

Dear Dr. Ahmed,

Thank you for submitting your manuscript to PLOS ONE. After careful consideration, we feel that it has merit but does not fully meet PLOS ONE’s publication criteria as it currently stands. Therefore, we invite you to submit a revised version of the manuscript that addresses the points raised during the review process.

We look forward to receiving your revised manuscript.

Kind regards,

Rajesh Kumar Singh, Ph.D.

Academic Editor

PLOS ONE

 [This study was funded by COMSTECH-NAPRECCA Consortium Through a  PhD Fellowship (award number: 3(250)/23-COMSTECH) awarded to KK at the International center for Chemical and Biological Sciences (ICCBS), University of Karachi.]. 

5. We note that your Data Availability Statement is currently as follows: [All relevant data are within the manuscript and its Supporting Information files.]

Additional Editor Comments:

The manuscript entitled "Green synthesized Polyscias fulva silver nanoparticles ameliorate Uterine Fibroids in Female Wistar Albino Rats" needs revision as per the suggestions of the reviewers.

Reviewers' comments:

Reviewer's Responses to Questions

**Comments to the Author**

1. Is the manuscript technically sound, and do the data support the conclusions?

Reviewer #1: Yes

Reviewer #2: No

2. Has the statistical analysis been performed appropriately and rigorously? 

Reviewer #1: Yes

Reviewer #2: No

3. Have the authors made all data underlying the findings in their manuscript fully available?

Reviewer #1: Yes

Reviewer #2: No

4. Is the manuscript presented in an intelligible fashion and written in standard English?

Reviewer #1: Yes

Reviewer #2: No

5. Review Comments to the Author

Reviewer #1: A good and comprehensive study addressing non surgical treatment for a common gynaecological problem affecting many women with different presentations however, i give the following minor comments:

-Abrrevations: i suggest to be included in a seperate section at the begining of the manuscript.

-Design of the study: should be stated in the methodology section as well as linked to the title.

Reviewer #2: Some of the suggestions that could improve the quality of this work include:

1. Polyscias fulva silver nanoparticles (PFAgNPs) reported the synthesis method without sufficient characterization of essential reaction conditions including temperature, reaction time and reducing agent that are key for reproducibility.

2. UV-visible spectroscopy shows maximum absorbance at 425 nm, but there is no comparison of this value with other silver nanoparticles formulations to confirm this observation.

3. Characterization of nanoparticle size by DLS (107.4 nm), XRD (25 nm) and SEM (35 nm) show discrepancies and lack of clarification about aggregation state or size variation under biological conditions.

4. The toxicity assessment does not mechanistically explain the lethargy, hyperventilation, and mortality at higher dosages, and does not explore markers of oxidative stress or inflammatory response.

5. The absence of a dose-response curve for the therapeutic effects hampers the identification of the relevant effective dose unaccompanied by adverse histological effects.

6. Sex hormone (estrogen and progesterone) and serum parameters (cholesterol, proteins) transduction effects need to be further described in terms of their biochemical mechanisms and possible connections to uterine fibroid regression.

7. The histological finding supported by morphometric determination or uterine tissue score in grade is not enough.

8. Discuss the few WaveFlex Biosensors in the literature survey to get interest in the readers. Advanced modalities like PFAgNPs may yet represent potential therapeutics, but studies need to consider long-term stability under physiological conditions.

9. The investigator has stated that without a control AgNP group (no Polyscias fulva present), it cannot be determined whether the observed effects are because of the plant extract or induced by the silver nanoparticles.

10. The limitations of extrapolation of Wistar rats to humans are discussed, but possible strategies, such as performing in vitro human cell studies to bridge this knowledge gap prior to a clinical trial, are not suggested.

6. PLOS authors have the option to publish the peer review history of their article (what does this mean? ). If published, this will include your full peer review and any attached files.

**Do you want your identity to be public for this peer review?** For information about this choice, including consent withdrawal, please see our Privacy Policy .

Reviewer #1: **Yes: ** Mohsen M A Abdelhafez

Reviewer #2: No

---

## [Author Response · Author response to Decision Letter 0]

22 Mar 2025

EDITORS COMMENTS

Response: The manuscript has been formatted as per PLOS ONE’s style requirements.

2. To comply with PLOS ONE submissions requirements, in your Methods section, please provide additional information regarding the experiments involving animals and ensure you have included details on (1) methods of sacrifice, (2) methods of anesthesia and/or analgesia, and (3) efforts to alleviate suffering. The (1) methods of sacrifice, (2) methods of anesthesia and/or analgesia, and (3) efforts to alleviate suffering have been included in the manuscript (Line 174-179)

In your Methods section, please provide additional information regarding the permits you obtained for the work. Please ensure you have included the full name of the authority that approved the field site access and, if no permits were required, a brief statement explaining why.; Permission to access the field sites was granted by the local area administration, as the plant specimens were collected from individual home gardens rather than from a protected forest area and hence no access permits were required. This has been included in the manuscript (Line 102-104).

 [This study was funded by COMSTECH-NAPRECCA Consortium Through a  PhD Fellowship (award number: 3(250)/23-COMSTECH) awarded to KK at the International center for Chemical and Biological Sciences (ICCBS), University of Karachi.]

Response: The funders had no role in study design, data collection and analysis, decision to publish, or preparation of the manuscript. This has been stated in the manuscript (Line 598-599).

5. We note that your Data Availability Statement is currently as follows: [All relevant data are within the manuscript and its Supporting Information files.]

Response: The data set has been deposited on this repository shared on this link: https://doi.org/10.6019/S-BSST1912 (Line 609-610)

Response: The data set has been deposited on this repository shared on this link; https://doi.org/10.6019/S-BSST1912(Line 609-610)

Reviewer 2

Comment 1: Polyscias fulva silver nanoparticles (PFAgNPs) reported the synthesis method without sufficient characterization of essential reaction conditions including temperature, reaction time and reducing agent that are key for reproducibility. Response 1: Thank you for the feedback. The reaction conditions, including temperature and reaction time, have now been explicitly stated in the manuscript (Lines 119-123) to enhance clarity and reproducibility. Regarding the reducing agent, this study did not use any external reducing agent. Instead, we highlighted in the discussion section that Polyscias fulva extract (PFE) itself possesses unique bioactive constituents capable of reducing silver ions to form PFAgNPs without the need for an additional reducing agent (Line 387-391). This intrinsic property of PFE has been elaborated as a novel aspect of our study, contributing to the green synthesis of silver nanoparticles.

Comment 2: UV-visible spectroscopy shows maximum absorbance at 425 nm, but there is no comparison of this value with other silver nanoparticles formulations to confirm this observation.

Response 2: In the discussion section (Lines 385-389), we have compared our findings with previous studies that have successfully demonstrated the formation of silver nanoparticles within the characteristic surface plasmon resonance (SPR) range of 400-450 nm. This comparison supports our observation that Polyscias fulva extract (PFE) facilitates the synthesis of silver nanoparticles without the need for an external reducing agent. We believe this strengthens the validity of our findings and highlights the unique potential of PFE in green nanoparticle synthesis. We appreciate the feedback. Characterization of nanoparticle size by DLS (107.4 nm), XRD (25 nm) and SEM (35 nm) show discrepancies and lack of clarification about aggregation state or size variation under biological conditions. Thank you for the observation. The differences in nanoparticle size obtained from DLS (107.4 nm), XRD (25 nm), and SEM (35 nm) are due to the inherent principles and measurement conditions of these techniques. DLS measures the hydrodynamic diameter of nanoparticles in suspension, which includes the nanoparticle core along with any surface-bound molecules and hydration layers, often resulting in larger size values. In contrast, XRD determines the crystalline domain size, which typically appears smaller as it does not account for nanoparticle aggregation or surface coatings. SEM provides the direct visualization of particle morphology and size in a dry state, where some degree of aggregation may occur, leading to intermediate size measurements.

To clarify the aggregation state and potential size variations under biological conditions, we have now included a discussion on the differences between these characterization methods and their implications in the revised manuscript. Additionally, we acknowledge that further stability studies under biological conditions could provide more insights into nanoparticle behavior, which we propose as a subject for future investigation. (Line 429-443)

Comment 3: The toxicity assessment does not mechanistically explain the lethargy, hyperventilation, and mortality at higher dosages, and does not explore markers of oxidative stress or inflammatory response.

Response 3: Thank you for the observation. We acknowledge that while our toxicity studies reported the observed effects, including lethargy, hyperventilation, and mortality at higher dosages, we did not delve into the precise mechanistic pathways underlying these responses.

To address this, we have now expanded the discussion to acknowledge the potential role of oxidative stress and inflammatory pathways in mediating these toxicological effects (Line 453-464).

Although specific biomarkers of oxidative stress (e.g., lipid peroxidation, glutathione depletion, ROS generation) and inflammatory responses (e.g., cytokine profiling) were not assessed in this study, we recognize their importance in providing mechanistic insights and have therefore recommended that future studies should investigate these markers to better elucidate the pathways involved in PFAgNP-induced toxicity. (Line 465-474)

Comment 4: The absence of a dose-response curve for the therapeutic effects hampers the identification of the relevant effective dose unaccompanied by adverse histological effects

Response 4: Thank you for the comment. We agree that the lack of a dose-response curve for the therapeutic effects limits identification of the optimal effective dose. While our study demonstrated the bioactivity of PFAgNPs, there is need for more detailed dose-response analysis that would provide a detailed understanding of the therapeutic window and help determine the lowest effective dose that does not induce adverse effects.

To address this, we have included this as a limitation in the discussion section and recommended future studies to include a comprehensive dose-response assessment. (Line 548-551)

Comment 5: Sex hormone (estrogen and progesterone) and serum parameters (cholesterol, proteins) transduction effects need to be further described in terms of their biochemical mechanisms and possible connections to uterine fibroid regression.

Response 5: We appreciate the comment. We have expanded the discussion to address the possible connections between hormonal modulation and serum parameter alterations. (Line 502-522)

Comment 6: The histological finding supported by morphometric determination or uterine tissue score in grade is not enough.

Response 6: Thank you for your insightful comment. We agree that while the histological findings in our study provided valuable information, a more comprehensive analysis involving morphometric determination or a uterine tissue grading score could further enhance the robustness of our conclusions. In light of your suggestion, we have now highlighted the need for incorporating detailed morphometric and grading techniques in future studies to provide a more comprehensive evaluation of tissue-level changes (Line 543-547).

Comment 7: Discuss the few WaveFlex Biosensors in the literature survey to get interest in the readers. Advanced modalities like PFAgNPs may yet represent potential therapeutics, but studies need to consider long-term stability under physiological conditions.

Response 7: Thank you for your thoughtful comment. We appreciate your suggestion to discuss the few WaveFlex Biosensors in the literature survey, as this could indeed generate greater interest among readers. In response, we have integrated this concept into the discussion section as a recommendation for future studies (Lines 552-565).

Comment 8: The investigator has stated that without a control AgNP group (no Polyscias fulva present), it cannot be determined whether the observed effects are because of the plant extract or induced by the silver nanoparticles.

Response 8: We appreciate the concern. However, we would like to clarify that the observed effects in this study are solely attributed to the Polyscias fulva silver nanoparticles (PFAgNPs). Prior to the efficacy studies, the PFAgNPs were thoroughly characterized using various techniques. These characterization results confirmed the successful synthesis and distinct physicochemical properties of the PFAgNPs that were tested, and thereby validate that the biological effects observed in the study are indeed due to the PFAgNPs.

Comment 9: The limitations of extrapolation of Wistar rats to humans are discussed, but possible strategies, such as performing in vitro human cell studies to bridge this knowledge gap prior to a clinical trial, are not suggested

Response 9: Thank you for your valuable comment. We agree that while the limitations of extrapolating data from Wistar rats to humans have been discussed, we did not explicitly propose strategies to bridge this knowledge gap before progressing to clinical trials. To address this, we have included the recommendation of the in vitro human cell studies and Advanced models such as 3D cultures or organ-on-a-chip system in the discussion section (Line 566-569).

---

## [Decision Letter · Decision Letter 1]

7 Apr 2025

Green synthesized Polyscias fulva silver nanoparticles ameliorate Uterine Fibroids in Female Wistar Albino Rats

PONE-D-24-56554R1

Dear Dr. Ahmed,

We’re pleased to inform you that your manuscript has been judged scientifically suitable for publication and will be formally accepted for publication once it meets all outstanding technical requirements.

Kind regards,

Rajesh Kumar Singh, Ph.D.

Academic Editor

PLOS ONE

Additional Editor Comments (optional):

Reviewers' comments:

Reviewer's Responses to Questions

**Comments to the Author**

1. If the authors have adequately addressed your comments raised in a previous round of review and you feel that this manuscript is now acceptable for publication, you may indicate that here to bypass the “Comments to the Author” section, enter your conflict of interest statement in the “Confidential to Editor” section, and submit your "Accept" recommendation.

Reviewer #1: All comments have been addressed

Reviewer #2: All comments have been addressed

2. Is the manuscript technically sound, and do the data support the conclusions?

Reviewer #1: Yes

Reviewer #2: Yes

3. Has the statistical analysis been performed appropriately and rigorously? 

Reviewer #1: Yes

Reviewer #2: Yes

4. Have the authors made all data underlying the findings in their manuscript fully available?

Reviewer #1: Yes

Reviewer #2: Yes

5. Is the manuscript presented in an intelligible fashion and written in standard English?

Reviewer #1: Yes

Reviewer #2: Yes

6. Review Comments to the Author

Reviewer #1: Thank you for the prompt response and appreciate addressing all the comments and wish you to continue research in same topic.

Reviewer #2: Authors have revised the manuscript properly. They have addressed the comments. Now, it is suitable for publication.

7. PLOS authors have the option to publish the peer review history of their article (what does this mean? ). If published, this will include your full peer review and any attached files.

**Do you want your identity to be public for this peer review?** For information about this choice, including consent withdrawal, please see our Privacy Policy .

Reviewer #1: **Yes: ** Mohsen M A Abdelhafez

Reviewer #2: No

---

## [Editor Report · Acceptance letter]

PONE-D-24-56554R1

PLOS ONE

Dear Dr. Ahmed,

I'm pleased to inform you that your manuscript has been deemed suitable for publication in PLOS ONE. Congratulations! Your manuscript is now being handed over to our production team.

Kind regards,

on behalf of

Dr. PLOS Manuscript Reassignment

Staff Editor

PLOS ONE